# Acne Vulgaris and Intake of Selected Dietary Nutrients—A Summary of Information

**DOI:** 10.3390/healthcare9060668

**Published:** 2021-06-03

**Authors:** Aleksandra Podgórska, Anna Puścion-Jakubik, Renata Markiewicz-Żukowska, Krystyna Joanna Gromkowska-Kępka, Katarzyna Socha

**Affiliations:** Department of Bromatology, Faculty of Pharmacy with the Division of Laboratory Medicine, Medical University of Białystok, Mickiewicza 2D Street, 15-222 Białystok, Poland; apodgorska1@student.umb.edu.pl (A.P.); renmar@poczta.onet.pl (R.M.-Ż.); krystyna.gromkowska.kepka@gmail.com (K.J.G.-K.); katarzyna.socha@umb.edu.pl (K.S.)

**Keywords:** acne, carbohydrates, fats, proteins, minerals, vitamins, prebiotics, probiotics, drinks, fibre

## Abstract

Acne vulgaris (AV) is a chronic disease that affects a significant percentage of the world’s population. Its development is influenced by both external and internal factors. The purpose of this review is to demonstrate the effect of basic nutrient intake on the exacerbation or alleviation of AV lesions. A retrospective review of publications in PubMed regarding diet therapy and the impact of individual nutrient intake on the skin condition of patients was conducted. Ingestion of products with a high glycaemic index may indirectly lead to sebum overproduction, which promotes infection with *Cutibacterium acnes* and causes inflammation. Consumption of certain dairy products may result in skin deterioration caused by the presence of hormones in these products, i.e., progesterone and testosterone precursors. The beneficial effect of fatty acids on the skin is manifested by the reduction in inflammation. Of significance in AV treatment are vitamins A, C, D, E and B, as well as mineral elements zinc and selenium. Proper nutrition may not only prevent or alleviate AV but also increase treatment efficacy.

## 1. Introduction

Acne vulgaris (AV) is one of the most common dermatological conditions, affecting approximately 15% of the world’s population. It is a chronic seborrhoeic disease and occurs primarily in young people during puberty. In girls, peak incidence occurs between 14 and 17 years of age, while in boys, the first symptoms are observed approximately 2 years later. During early adulthood, the prevalence of AV decreases spontaneously. However, it may appear in older persons as so-called ‘adult-onset acne’. Incidence rates of AV for males are different from those of females. More severe forms of the disease are observed in men. AV affects mainly seborrheic zones, primarily those located on the face (99%). It also occurs on the back (90%) and chest (70%). It is a disease associated with hair follicles and sebaceous glands, characterised by the presence of open and closed comedones, papules, pustules or purulent cysts [1,2].

AV is a chronic disease of the sebaceous glands, accompanied by seborrhoea. Sebaceous hyperplasia and the tendency of the sebaceous glands to excessive sebum production, keratosis of hair follicles, bacterial infections (e.g., *Cutibacterium acnes*) or hormonal activity contribute to disease development. In addition, both extrinsic and intrinsic factors affect the course of AV. The main external factors which exacerbate AV include an improper diet rich in highly processed foods, incorrect skin care and environmental factors such as air pollution. The key intrinsic factors are hormonal changes and genetic predisposition [1,2,3].

The aim of AV treatment is to prevent excessive keratosis, reduce seborrhoea and inhibit the growth of *Cutibacterium acnes*. The treatment regimen involves administration of both local treatments and antibiotics (erythromycin, clindamycin, tetracyclines and macrolides), retinoids (tretinoin, isotretinoin, adapalene and tazarotene), benzoyl peroxide or azelaic acid. A properly balanced diet also plays an important role in relieving AV symptoms [2].

AV has been the subject of many publications over the past 20 years as shown in Figure 1. Therefore, the aim of this review was to collect and demonstrate the effect of the intake of essential nutrients on the development and course of AV.

## 2. Materials and Methods

This article provides an overview of the consumption of individual nutrients and their beneficial effects on AV treatment. Research on the importance of nutrition in AV published since 2005 was reviewed. The search was carried out in PubMed. The following terms were entered: ‘acne’ (criterion: word in the title of the publication) and ‘carbohydrates’, ‘proteins’, ‘fats’, ‘vitamins’, ‘minerals’, ‘probiotics’, ‘drinks’, alcohol’, ‘spices’, ‘fibre’ and ‘prebiotics’. The inclusion criterion were English language and full-text articles. In the second stage, publications unrelated to the topic discussed (including those concerning the external use of ingredients or the use of other methods of therapy) were rejected. In addition, the discussion included publications on topics that were not found in the database during the literature review, in accordance with the planned criteria.

## 3. Results

Figure 2, according to [4] with the authors’ own modification, shows the results of the search for the selected terms. Most publications were excluded because they discussed other aspects of AV therapy (pharmacological treatment, cosmetology and reference to other medical conditions).

The influence of selected nutritional factors on AV described in publications found to be in accordance with the assumed criteria is presented in Table 1. In addition, Table 2 summarises the studies that were found during the database review to discuss and explain the role of nutritional factors but were not displayed according to the assumed criteria for the literature review.

## 4. Discussion

### 4.1. Carbohydrates

Carbohydrates play a significant role in the pathogenesis of AV and may exacerbate its course. A study by Akpinar Kara et al. (2019) demonstrated a positive correlation between carbohydrate intake and the severity of AV assessed on the Global Acne Grading System (GAGS) scale [24]. It was associated with an increase in glucose level depending on the Glycaemic Index (GI), i.e., a measure indicating the rate of increase in blood sugar level after consuming 50 g of carbohydrates in reference to blood glucose level after ingesting 50 g of pure glucose [35].

A diet rich in carbohydrates with a high GI is associated with the occurrence of hyperglycaemia, hyperinsulinemia and increased production of insulin-like growth factor 1 (IGF-1) as well as development of insulin resistance [36]. Insulin may affect the function of the liver and the adrenal and pituitary glands, thus promoting the production of, among others, androgen and sex hormone binding globulin (SHBG) as well as participating in sebum production. The glycaemic load (GL) is also important, as it is a measure that takes into account the GI and carbohydrate content in a portion of food. This was confirmed in a group of men aged 17 ± 4 years, some of whom (*n* = 5) consumed 45% of their calories from carbohydrates, while some (*n* = 5) consumed 55% of their calories from carbohydrates. It was demonstrated that SHBG levels were significantly lower in the group with higher carbohydrate consumption, indicating that a rise in GL may increase IGF-1 and sex steroid levels. Therefore, low SHBG concentrations may be the cause of high levels of androgens in their free form in the blood, which increases sebum secretion by the sebaceous glands [20,35].

A study by Reynolds et al. (2010) showed that a diet with a low GI (51 ± 1), compared with a diet with a high GI (62 ± 2), enhanced the appearance of the skin of AV patients, but it was not a statistically significant change. The authors suggest that observations longer than 8 weeks and greater reductions in the glycaemic index are necessary to confirm improvement in AV in adolescent boys [15].

A 10 week dietary intervention was conducted which involved adherence to a diet with a low GL, rich in vegetables, fruit, beans, fish, barley and whole wheat bread. Histopathological skin examination revealed a reduction in inflammation, a decrease in the size of the sebaceous glands as well as lower expression of sterol-regulating element-binding protein-1 and interleukin-8. An increase in IGF-1 levels contributes to an increase in the bioavailability of androgens, enhanced growth of skin cells and, thus, clogging of hair follicles and formation of comedones. These factors also stimulate the growth of sebaceous gland keratinocytes, causing abnormal peeling of dead epidermal cells, corneocytes and sebum overproduction, which promotes inflammation induced by *Cutibacterium acnes* [12].

The above theory was also confirmed in a study conducted on a group of 66 people, 34 of whom followed a low GI and GL diet but only for 2 weeks. However, it allowed to notice a beneficial clinical effect [5].

A study conducted among 57 patients with AV showed that they consumed more chocolate than the control group. No significant relationship was found between the consumption of other sweets, potatoes, chips or carbonated drinks, and AV severity was assessed according to the Comprehensive Acne Severity Scale (CASS) [23].

The effect of a low glycaemic load (LDL) diet was confirmed in three other studies [16,17,18]. The interventions lasted for 12 months.

Food products can be divided into three groups: those with a low (≤55), medium (55–69) and high (≥70) GI. People with AV should choose products with a low GI such as grapefruit (GI: 25), cooked chickpeas (28 ± 6), apples (38 ± 2), baked potatoes (41), cooked buckwheat (45) and oatmeal bran bread (47 ± 3). Low GL products such as strawberries (GL: 1), watermelon (4), broad beans (9), boiled corn (9) should also be included in the diet [37].

### 4.2. Fibre

Fibre also belongs to the carbohydrate group. Its role in supporting AV therapy has not been sufficiently elucidated to date. In a study conducted on a group of men aged 15–25 years who were recommended to maintain a low GI diet, an increase in the average fibre content in the diet (from 25.3 ± 1.8 to 36.9 ± 2.0 g) was noted after diet cessation while fibre consumption in the control group did not change (before the study: 25.2 ± 2.1, after the study: 25.2 ± 2.0). Individuals who followed a low GI diet for 12 weeks showed a greater reduction in AV lesion counts [18].

Fibre rich foods include vegetables (e.g., white beans), fruit (e.g., dried plums) and cereals (e.g., oat flakes) [38].

### 4.3. Proteins

One of the main nutritional sources of protein is dairy, particularly cow’s milk and its products. They contain casein (constituting approximately 80% of total protein) and whey proteins (constituting 20–25% of cow’s milk proteins, of which approximately 75% are albumins including α-lactalbumin and β-lactalbumin). Proteins with antimicrobial, antifungal, antibacterial and antiviral activity, such as lactoferrin, are also found in milk [39]. Moreover, milk products contain many bioactive compounds, e.g., hormone precursors, IGF-1 or transforming growth factor beta (TGF-β) [40].

Consumption of milk has the same health consequences as the intake of high GI products described above. Milk is also a high GI product because of the presence of branched chain amino acids (BCAAs), leucine, isoleucine and valine, stimulating the secretion of insulin in the pancreas, which increases insulin and IGF-1 levels in the blood. Furthermore, it contributes to the growth of the sebaceous glands and lipogenesis [36,40].

Steroid compounds present in milk (i.e., androgens, reduced forms of steroids or non-steroidal growth factors) contribute to the comedogenicity of milk. Consumption of skimmed milk has a particularly negative effect on the functioning of sebaceous glands. Androgens can cause greasy skin, development of *Cutibacterium acnes* in hair follicles or excessive keratosis as confirmed in a study conducted on a group of girls aged 9–15 years (*n* = 6094) [22].

The condition of skin affected by AV is also influenced by whey proteins, primarily α-lactalbumin. They are characterised by the presence of leucine (14%), as a result of which they affect epidermal proliferation, androgen synthesis and lipogenesis in the sebaceous glands [36]. Whey proteins are used in the form of concentrates or isolates in, among others, protein supplements. They have a higher insulin index in comparison with cow’s milk, which results from the process of removing fat from the product and isolating and increasing the number of individual exogenous amino acids in it. AV lesions have been observed in athletes and bodybuilders who use whey and casein-rich protein supplements. A study conducted on a group of young men with AV, who were practicing bodybuilding and taking whey protein, showed that discontinuing protein supplementation significantly improved their skin condition [25].

A positive correlation between the consumption of whole and skimmed milk and the severity of AV lesions was confirmed in a retrospective study conducted on a group of 47,355 women. Deterioration in skin condition was also caused by the consumption of other dairy products such as cottage cheese, cream cheese and milk drinks [21]. It was associated with the presence of hormones in these products (i.e., progesterone, the testosterone precursors—androstenedione and dehydroepiandrosterone sulfate—or reduced steroids, which are precursors of dehydrotestosterone (DHT)), as well as the presence of biologically active substances including glucocorticoids or IGF-1 [41].

The relationship between milk consumption and occurrence of AV was proven in a large prospective study (*n* = 6094) [22]. Moreover, other studies showed significantly higher consumption of milk and chocolate [23] and cheese [24] in patients with AV. A relationship has also been demonstrated between whey protein consumption and AV exacerbation [25].

### 4.4. Fats

A beneficial effect on skin condition is exerted by polyunsaturated fatty acids (PUFAs), such as omega-3 and -6, including essential unsaturated fatty acids (EFAs), i.e., linoleic acid (LA) and α-linolenic acid (ALA). The human body cannot synthesise these acids and, therefore, they must be supplied with food. Vegetable oil (e.g., sunflower, linseed, soybean, grapeseed, peanut or sesame oil) are rich sources of LA, whereas chia seeds, walnuts, olive oil and rapeseed oil provide ALA. In addition to EFAs, PUFAs also include γ-linolenic acid (GLA), arachidonic (AA), eicosapentaenoic acid (EPA) and decozahexaenoic acid (DHA). GLA is found in hemp seed oil and blackcurrant seeds, while EPA and DHA can be found mainly in marine fish fat (mackerel, salmon, cod, herring) [38,42].

Studies conducted by Jung et al. (2014) demonstrated that omega-3 fatty acids (ALA, DHA, EPA) have a beneficial effect on skin condition and reduce the incidence of AV. The study group ingested 2000 mg of EPA and DHA (*n* = 15) or 1000 mg of borage oil containing 400 mg of γ-linolenic acid (*n* = 15) for 10 weeks. Using eosin and haematoxylin staining, a reduction in non-inflammatory and inflammatory AV lesions was demonstrated [9]. Inflammation is reduced by inhibiting the production of proinflammatory cytokines and leukotriene B4 [43]. Study results indicated that despite the side effects observed, such as transient diarrhoea (6.7%) and mild gastrointestinal discomfort (6.7%), the above supplementation may be used as an adjuvant therapy [9].

EPA, found in fish oil, blocks the conversion of arachidonic acid to leukotriene B, a pro-inflammatory factor that increases the production of sebum. Moreover, omega-3 fatty acids liquefy sebum and increase skin tolerance to bacterial agents. Omega-6 acids have the opposite effect to that of omega-3 acids. They belong to a group of precursors of inflammatory factors. Increasing the supply of omega-6 fatty acids leads to the development of inflammatory factors and, thus, the formation of AV [9]. One of the reasons for the thickening of sebum, which leads to the obstruction of the sebaceous glands as well as excessive keratosis resulting in comedones, is a deficiency of PUFAs in the diet. It also changes the pH of the skin, leading to colonisation of bacteria and fungi that cause skin inflammation [44,45]. Saturated fatty acids (e.g., palmitic, stearic or myristic acid), including trans isomeric acids, the main sources of which are hydrogenated vegetable oils found in margarine, confectionery or fast food, have a negative effect on AV-affected skin [46]. High content of palmitic acid in the diet leads, through its influence on IL-1β and IL-1α, to the development of skin inflammation, enhances comedogenesis and increases sebum secretion. Diet-induced metabolic changes give rise to AV-related inflammation. Omega-3 PUFA is a promising component of supplements supporting acne therapy [41].

### 4.5. Vitamins

Vitamins important for the skin condition of AV patients include A, C, D, E and B vitamins.

Vitamin A impacts the formation of new cells and accelerates their regenerative processes. Retinoids, vitamin A derivatives, are used in the treatment of AV. They are lipophilic molecules that easily penetrate the epidermis. By binding to nuclear receptors, retinoids can modulate gene expression, including genes involved in cell proliferation. Furthermore, they exert a beneficial effect on the processes of synthesis and degradation of collagen and hyaluronic acid. Retinoids, thanks to their ability to capture free radicals and absorb UV radiation, have a protective effect on skin aging. Hence, vitamin A and retinoids are components of many drugs, dietary supplements and dermocosmetics including those that support the treatment of AV-affected skin [47].

Sources of retinol are butter, fish and calves’ liver. Sources of beta-carotene, a provitamin A carotenoid, include carrots, pumpkin, peppers, apricots, melon and papaya [38].

Due to the number of health benefits, products with high vitamin C content should be part of the diet of AV patients. However, vitamin C is more available when administered topically rather than systemically as demonstrated in a three-month study by Traikovich (1999) [48]. This vitamin has an anti-inflammatory effect because of its ability to inhibit NF-kB, which is responsible for the activation of inflammatory cytokines including IL-1, IL-6, IL-8 and TNF-alpha. Furthermore, its wound healing and anti-hyperpigmentation properties as well as antioxidant properties are important in the skin care of AV patients [48,49].

Foods rich in vitamin C include blackcurrants, peppers, guava, parsley, wild rose and citrus fruit [38].

Vitamin D has a pleiotropic effect by preventing AV lesions via inhibition of cell division, decrease in sebum secretion, prevention of pore blockage and inhibition of *Cutibacterium acnes* growth. A beneficial influence of vitamin D on the skin of AV patients was demonstrated in a study of 100 AV patients. For three months, 0.25 μg alfacalcidol supplementation was administered. Significantly higher vitamin D serum levels were observed in AV patients receiving the supplementation. A decrease in IL-6 and TNF-alpha levels was also demonstrated [26]. Lim et al. (2016) measured the level of 25 (OH) D in a group of 160 patients and demonstrated that among people with AV, as many as 48.8%, had abnormal levels of this vitamin. Importantly, supplementation with 1000 IU/day for 2 months allowed for a significant reduction in the inflammation accompanying AV [7].

Fatty fish, fish oil, liver, eggs and yeast are excellent sources of vitamin D [38]. This vitamin can also be synthesised when the skin is exposed to solar radiation for 10–15 min, two to three times a week. In the course of a photochemical reaction, 7-dehydrocholesterol is transformed into vitamin D3 under the influence of ultraviolet B (UVB) with a wavelength of 280–320 nm [50].

Vitamin E, which is a component of the skin’s lipid mantle responsible for skin hydration, also helps in the treatment of AV. This vitamin has a strong antioxidant, anti-inflammatory and anti-seborrheic effect. A decrease in vitamin E level, manifested by hyperkeratosis and dry skin, is observed in AV patients. Studies in persons 28.54 ± 8.3 years of age demonstrated that plasma concentrations of vitamin E were significantly lower in patients with AV (7.88 ± 3.0 μmol/L) compared to the control group (11.06 ± 3.1 μmol/L) [28].

Data from the literature also indicate the possibility of using, inter alia, vitamins to relieve the side effects of acne treatments. For example, vitamins with antioxidant properties (vitamin C and vitamin E) in combination with other ingredients, such as gamma linolenic acid, beta-carotene, coenzyme Q10 or *Vitis vinifera*, used in patients treated with isotretinoin, reduced the side effects of treatment with this drug, including reducing dryness and redness of the skin [8]. Vitamin E alone does not show such an effect [19].

Vitamin E is found primarily in eggs, green leaves of vegetables, tomatoes, walnuts and oils [38].

B vitamins useful in the treatment of AV include vitamin B1, B2, B3, B5, B6 an B7. Their action is mainly based on inhibiting sebum secretion and reducing colonisation of *Cutibacterium acnes*. A study on a group of 41 individuals with AV revealed that 12 week supplementation with pantothenic acid (two tablets twice a day containing a total of 4.4 g of the substance) resulted in a reduction of inflammation as well as an increase in the Dermatology Life Quality Index (DLQI) [27].

Nutritional sources of B vitamins include yeast, nuts, oatmeal, fish, lean meat, bran or liver [38].

On the other hand, dietary supplementation with high doses of vitamin B12 is suspected of worsening AV lesions, especially in women. Taking high doses of vitamin B12 weekly (> 5–10 mg) or using it for a longer period of time may also have an influence on the appearance of AV [51,52]. Balta and Ozugus (2014) [51] described in their work a case report of a 37-year-old woman with AV-like eruptions after intramuscular injection of vitamin B12 (two days, 1000 µg of vitamin B12). The lesions appeared on the chest, back, face and neck within 12 h after the second dose of the vitamin. Dermatological tests revealed maculopapular eruptions on the face (with the forehead being the most affected) upper back and chest but without comedones or cysts. Bacteriological and mycological cultures of the lesions gave negative results. Based on the research, the patient was diagnosed with vitamin B12-induced AV-like eruptions [51]. Velardi et al. (2018) described the case of five women, aged 37, 32, 62, 29 and 21 years old, who developed AV-like lesions caused by vitamin B12. During intramuscular or oral vitamin B12 therapy, within 1 week to 5 months, the patients began to develop facial lesions in the form of papules and pustules, three of them also on the neck, upper chest, back and arms. No cysts or blackheads were seen. Two women had very high vitamin B12 serum levels, while in one patient eosinophilic folliculitis was identified during histopathological examination. Spontaneous remission of lesions was noted at 3 to 6 weeks after discontinuation of vitamin B12 therapy [52].

### 4.6. Minerals

Zinc and selenium are the most important minerals that support AV therapy. Zinc has a bacteriostatic effect on *Cutibacterium acnes*, inhibits chemotaxis and reduces production of inflammatory cytokines. This element has antioxidant and anti-inflammatory properties. A study by Ozuguz et al. (2014) revealed that AV patients displayed significantly lower serum concentrations of zinc (62.15 ± 18.1 μmol/L) compared to the control group (81.57 ± 20.4 μmol/L) [28].

Sources of zinc include oysters, pumpkin and sunflower seeds and whole grains [38].

Selenium regulates sebum production and has anti-inflammatory properties. In AV therapy, it is usually used together with vitamin E and zinc due to the fact of their antioxidant properties. Oxidative stress is implicated in the pathophysiology of AV and, therefore, attempts to combine selenium (200 μg/day) with silymarin (70 mg, 3 times/day) and N-acetylcysteine (600 mg, 2 times/day) have been made. As a result of such supplementation, a decrease in the levels of malonic aldehyde and IL-8 and an increase in the level of glutathione was demonstrated in patients with AV, reducing the number of inflammatory lesions [29].

A visible reduction in inflammatory lesions was also reported in patients with AV after the combination of copper, zinc, nicotinamide, azelaic acid, pyridoxine and folic acid [11].

Brazil nuts are the main source of selenium, but it is also present in fish and red meat [38].

### 4.7. Probiotics and Prebiotics

In recent years, the importance of the gut–skin axis has been emphasised. The intestinal microbiome has a considerable influence on the functioning of the body. Its disorders are involved in the pathogenesis of allergic, cardiovascular, gastrointestinal, mental and dermatological diseases. Proper supplementation prevents colonisation of pathological bacterial flora.

Probiotics, as defined by experts at the Food and Agriculture Organization of the United Nations (FAO) and the World Health Organization (WHO), are live microorganisms that—when administered in appropriate amounts—provide a health benefit to the host [53].

Prebiotics are an indigestible food component that stimulates the growth or activity of bacteria present in the colon.

Their consumption has a beneficial effect in the treatment of AV. The purpose of Jung et al. (2013) was to assess the effect of probiotics on reducing the side effects of systemic antibiotic (minocycline) and their synergistic action with antibiotic in the treatment of AV. The study involved 45 women aged 18–35, who were randomly assigned to one of three groups. Group A comprised women using probiotics, group B those using antibiotic and group C consisted of women treated with both probiotics and antibiotic. Clinical assessments were made at the start of the study and at 2, 4, 8 and 12 weeks at follow-up visits. After 4 weeks, all women showed a significant improvement, while after 8 and 12 weeks, group C showed a significant decrease in the number of AV lesions, compared to the other two groups [10].

Rahmayani et al. (2019) assessed the effect of probiotics on serum IL-10 levels in AV as well as their side effects. The study group consisted of 33 people with AV whose serum levels of IL-10 were measured before and after probiotic supplementation for 30 days. Increased serum levels of IL-10 were demonstrated in people with AV. The side effect of using the probiotics was flatulence, which was found in only two people in the first week of the study, but this is a tolerable side effect [30].

It has been shown that in women with AV, supplementation with fructo-oligosaccharides (FOS) and galacto-oligosaccharides (GOS) for 3 months reduces insulin and C-peptide levels, which confirms the effect on glycaemic parameters, particularly important in pathogenesis AV. The literature reports indicate a relationship between AV, high glycaemic index and metabolic syndrome [31].

### 4.8. Other Important Ingredients in the Diet

Proteins, fats, carbohydrates, vitamins and minerals are components of a well-balanced meal. The influence of green tea, alcohol and hot spices on the skin condition of patients with AV should also be emphasised.

In the diet of individuals with AV, milk, protein supplements and other dairy products should be limited or completely eliminated. On the other hand, consumption of fermented dairy products (e.g., kefir or buttermilk) is recommended due to the presence of lactoferrin and *Lactobacillus* bacteria in these foods. These ingredients reduce skin inflammation and decrease sebum production. In addition, lactoferrin inhibits the proliferation of *Cutibacterium acnes* by damaging their cell membranes and, thus, leading to their death. Kim et al. (2010) assessed the effect of lactoferrin-enriched fermented milk on skin surface lipids and clinical improvement in AV patients. Lactoferrin is a protein of whey milk isolated after removal of precipitated casein. It has anti-inflammatory and systemic effects in inflammatory diseases. The study group consisted of 36 patients aged 18–30 years, who were randomly divided into two groups: the lactoferrin group (*n* = 18) and the placebo group (*n* = 18). For 12 weeks, the patients were fed daily with fermented milk containing 200 mg of lactoferrin, after which the amount and severity of common AV were assessed during monthly visits. Moreover, the level of skin hydration, sebum secretion, pH and the amount of lipids on the skin’s surface were assessed, both before the test and after 12 weeks. Patients in the lactoferrin group manifested a significant reduction in inflammatory lesions (approximately 39%) and AV severity (approximately 20%) after 12 weeks compared to the placebo group. In this group, the amount of secreted sebum also decreased (by approximately 31%). The presence of lipids on the skin surface in both groups was comparable, but only in the lactoferrin group was the content of triacylglycerols reduced, which is associated with a decrease in the amount of sebum, the number of AV lesions and their severity [14].

Lactoferrin (100 mg) has a beneficial effect both in combination with vitamin E (11 IU as alpha-tocopherol) and zinc (5 mg as zinc gluconate), taken twice daily [6], as well as alone as 100 mg of lactoferrin-enriched (80%) whey milk protein powder, twice a day [13] and in the amount of 200 mg of lactoferrin alone—as described above [14].

Green tea is one of the substances that reduce the number of inflammatory lesions on the nose and chin. This was confirmed in a study including 80 women aged 25–45 years. Participants with AV consumed a decaffeinated green tea extract containing 856 mg epicatechin gallate. Beneficial effects were demonstrated in 80% of study participants [34].

Among other dietary factors which exacerbate AV lesions, alcohol consumption should be mentioned. This was confirmed in the study by Suh et al. (2011). Nutritional, behavioural, clinical and epidemiological factors were analysed in a group of 1236 patients from 17 hospitals. Alcohol consumption was shown to be more strongly correlated with severity of AV symptoms in men than women [54].

Spicy and salty foods are thought to increase the incidence of AV. To investigate this problem, El Darouti et al. (2016) analysed 200 patients with AV. A negative correlation was found between the amount of sodium chloride consumed and the age of patients affected by AV (*r* = −0.216, *p* = 0.031). AV patients consumed higher amounts of the tested ingredient (median 3367.54 mg) compared to controls (median 2271.8 mg). However, consumption of spicy foods did not correlate with disease severity or duration [33].

Consumption of sweet drinks and the occurrence of AV was the subject of research by Huang et al. (2019) [32]. The study was conducted on a group of 8197 students. It was demonstrated that frequent consumption of carbonated drinks, fruit-flavoured drinks and sweetened tea (more than seven times a week, particularly consumption of more than 100 g sugar) was associated with moderate or severe AV.

## 5. Conclusions

Diet largely determines the condition of the skin. Appropriate food products and supplementation, providing primarily vitamin D, omega-3 fatty acids, vitamins and minerals with antioxidant properties, prebiotics and probiotics, and additional use of green tea extract, as well as reducing the consumption of milk, salty and spicy foods and products with a high glycaemic index may not only prevent AV and alleviate lesions but also facilitate treatment of the condition.

## Figures and Tables

**Figure 1 healthcare-09-00668-f001:**
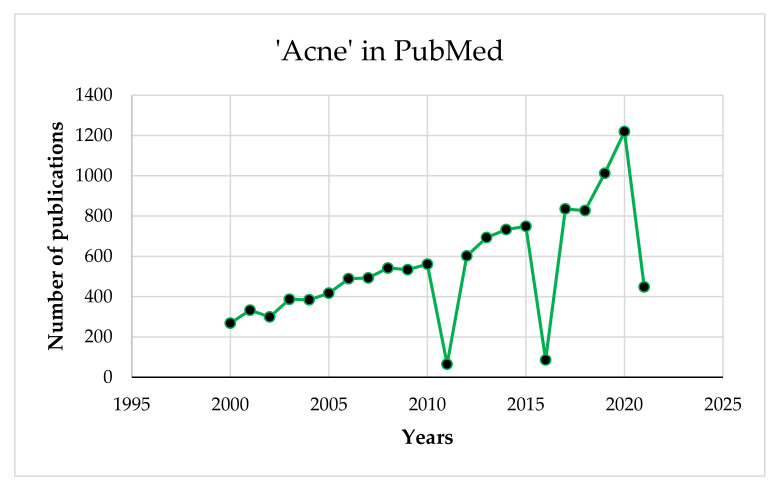
PubMed search results for ‘acne’ from 2000 to 2021.

**Figure 2 healthcare-09-00668-f002:**
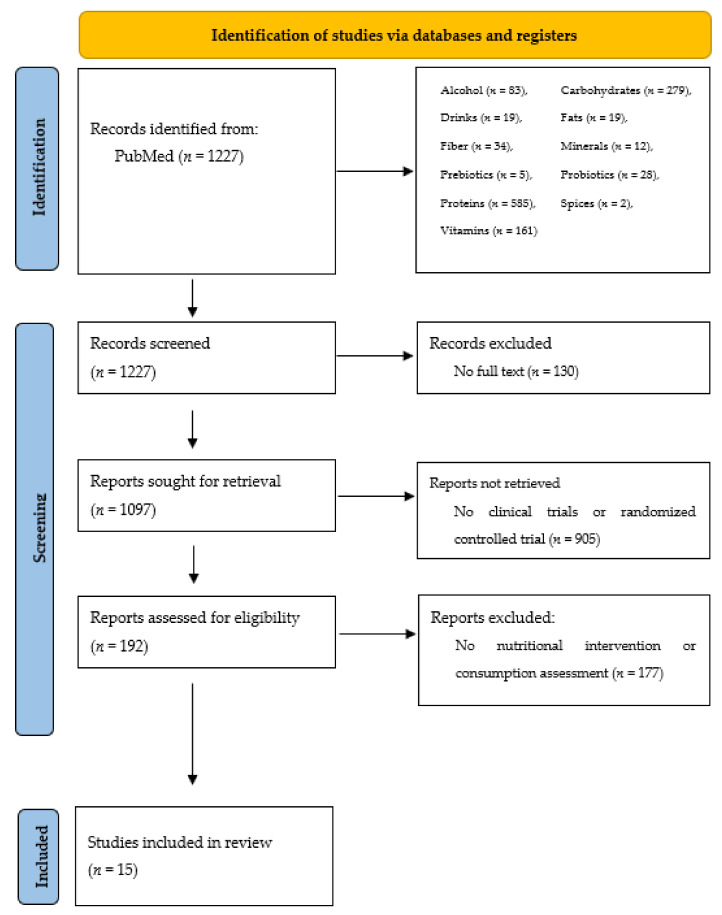
Literature review results, including exclusion factors.

**Table 1 healthcare-09-00668-t001:** Characteristics of studies describing acne diet therapy.

*n*	Type of Acne	Type of Study	Intervention/Measurement	Duration	Effect	Year [Reference]
66	moderate to severe AV	randomised controlled trial	-group 1: *n* = 34–participants with low GI and GL diet -group 2: *n* = 32–participants with usual eating diet	2 weeks	-IGF-1 concentrations significantly decreased among the group with low GI and GL diets and control group -there were no significant differences in insulin, glucose, IGFBP-3 concentrations and insulin resistance	2018 [5]
168	mild to moderate AV	randomised, double-blind, placebo-controlled trial	-group A (*n* = 82): capsules containing lactoferrin (100 mg), vitamin E (11 IU, as alpha-tocopherol), and zinc (5 mg, as zinc gluconate), capsule twice a day -group B (*n* = 82): placebo, capsules containing starch, capsule twice a day	12 weeks	-in the lactoferrin group, compared to the control: significant reduction in the total number of lesions (already after 2 weeks, with a maximum: at week 10), reduction in comedones and inflammatory changes (at week 10), sebum (improvement by week 12)	2017 [6]
160	nd	case-control study with a randomised controlled trial	*n* = 80 patients *n* = 80 healthy controls 1st stage: 25 (OH) D deficiency in 48.8% of AV patients, but only 22.5% of healthy controls 2nd stage: supplementation of 1000 IU of cholecalciferol/day	2 months	-deficiency of vitamin D was more frequent in patents with AV, oral vitamin D supplementation significantly improved AV inflammation	2016 [7]
48	nodular acne	randomised controlled trial	*n* = 24 patients received isotretinoin therapy (20–30 mg/day) and dietary supplement (gamma linolenic acid, vitamin E, vitamin C, beta-carotene, coenzyme Q10 and *Vitis vitifera*, twice a day) *n* = 24 patients received only isotretinoin (20–30 mg/day)	6 months	-patients using a dietary supplement with antioxidant properties had fewer side effects resulting from the use of isotretinoin, less redness and dryness and a better degree of hydration	2014 [8]
45	mild to moderate AV	randomised controlled trial	-group I: 2000 mg of eicosapentaenoic acid and docosahexaenoic acid) -group II: borage oil containing 400 mg γ-linoleic acid),III: a control group	10 weeks	-subjective assessment of improvement -reduction in inflammation and non-inflammatory acne lesions	2014 [9]
45	mild to moderate AV	prospective, randomised, open-label trial	-group A: probiotic -group B: minocycline -group C: probiotic and minocycline	12 weeks	-a significant improvement in the total number of lesions 4 weeks after the start of treatment in all groups -after 8 and 12 weeks, group C had a significant decrease in the total number of lesions compared to groups A and B	2013 [10]
235	inflammatory AV	multi-centre, open-label, prospective study	-addition of 1 to 4 tablets NicAzel (nicotinamide, azelaic acid, zinc, pyridoxine, copper, folic acid) to the current acne treatment regimen	8 weeks	-visible reduction in inflammatory changes (88% of patients), and 81% of patients rated skin appearance as moderate or much better	2012 [11]
32	mild to moderate AV	randomised, controlled trial	*n* = 17–low-glycaemic-load diet *n* = 15–control group	10 weeks	-significant clinical improvement in the number of non-inflammatory and inflammatory AV lesions (in the low-glycaemic index group)	2012 [12]
43	mild to moderate	exploratory study	100 mg of lactoferrin-enriched (80%) whey milk protein powder, twice a day	8 weeks	-reduction in inflammatory lesions (20.2%), non-inflammatory lesions (23.5%) and total lesions (22.5%), reduction in total lesion count (76.9%)	2011 [13]
36	mild to moderate	double-blind, placebo-controlled study	*n* = 18–lactoferrin group (200 mg daily) *n* = 18–placebo group	12 weeks	-a significant reduction in acne severity (by 20.3%), number of inflammatory lesions (by 38.6%) and total lesions (by 23.1%) compared to the placebo group	2010 [14]
58	1 (mild), 2 (moderate), or 3 (severe)	controlled clinical trial	*n* = 23–low glycaemic index *n* = 20–high glycaemic index	8 weeks	-improvement in the appearance of the skin (but not statistically significant), change in insulin sensitivity (but not statistically significant)	2010 [15]
31	mild to moderate AV	dietary intervention trial	*n* = 16, LGL group *n* = 15–control group	12 weeks	-increases in the ratio of saturated to monounsaturated fatty acids of skin surface triglycerides compared to controls	2008 [16]
43	mild to moderate	randomised, investigator-masked, controlled trial	*n* = 23–LGL group, *n* = 20–control group	12 weeks	-decrease in the total number of lesions, body weight, free androgen index and increase in insulin-like growth factor-1 binding protein compared to the control group	2007 [17]
43	mild to moderate	non-randomised, parallel, controlled feeding trial	*n* = 23, patients: LGL diet (25% energy from proteins, 45% energy from carbohydrates) *n* = 20, control group: diet rich with carbohydrates, but without reference to the glycaemic index	12 weeks	-decrease in the total number of lesions, decrease in body weight and body mass index, greater improvement in insulin sensitivity-compared to the control group	2007 [18]
82	severe AV, moderate AV unresponsive to conventional therapy, scarring AV, and AV causing psychological disorder	investigator-blinded, randomised study	group 1: isotretinoin (1 mg/kg/day) group 2: isotretinoin (1 mg/kg/day) and vitamin E (800 IU/day)	16 weeks	-vitamin E did not reduce the side effects associated with the use of isotretinoin	2005 [19]

GI—glycaemic index, GL—glycaemic load and LGL—low glycaemic load.

**Table 2 healthcare-09-00668-t002:** Characteristics of publications on topics that were not found in the database during the literature review.

*n*	Type of Study	Intervention/Measurement/Methods	Duration	Effect	Year [Reference]
				Carbohydrates	
12	randomised, controlled trial	low glycaemic load diet (LGL) or high glycaemic load diet (HGL)	12 weeks	-changes in the homeostasis model assessment of insulin resistance (HOMA-IR): −0.57 for LGL vs. 0.14 for HGL -changes in sex hormone binding globulin (SHBG): SHBG levels decreased significantly from baseline in the HGL group -changes in binding proteins (IGFBP-I and IGFBP-3): IGFBP-I and IGFBP-3 levels increased in the LGL group	2008 [20]
				Proteins	
47,355	prospective cohort study	assessment of the association of milk consumption with the occurrence of AV	questionnaires on high school diet	a positive relationship between the consumption of whole and skimmed milk and the incidence of AV	2005 [21]
6094	prospective cohort study	association of correlation between drinking milk and AV	nd	-positive association between the intake of milk and AV	2006 [22]
114	case-control study	intake assessment–food intake questionnaire *n* = 57 patients *n* = 57 control group	nd	-milk and chocolate consumption were significantly higher in patients with AV	2018 [23]
106	case-control study	dietary intake of milk and dairy products along with carbohydrate/ fat/protein ratios *n* = 53 patients *n* = 53 control group	3 day (2 weekdays and 1 weekend day) consumption record	-statistically higher consumption of cheese in people with AV	2019 [24]
5	case report	developing AV after the consumption of whey protein	5.6 ± 1.8 months	-milk and dairy products were enhancers of insulin/insulin-like growth factor 1 signalling and AV aggravation	2012 [25]
				Vitamins	
200	prospective, randomised, controlled and open label trial	investigated the serum level of 25-hydroxy vitamin D in AV patients and assessment of the efficacy and safety of active vitamin D in management of AV *n* = 100 patients *n* = 100 control group	3 months	-serum levels of 25-hydroxy-vitamin D were lower in AV patients than in healthy controls -AV patients were more likely to have a vitamin D deficiency than healthy people	2020 [26]
48	randomised, double-blind, placebo-controlled study	determination of the safety, tolerability and effectiveness of daily administration of an orally administered pantothenic acid-based dietary supplement in men and women with facial AV lesions *n* = 23 patients, *n* = 25 placebo group	12 weeks	-reduction in total lesion count in the pantothenic acid group versus placebo -reduction in inflammatory lesions was significantly reduced -dermatology life quality index (DLQI) values were lower at week 12 in the pantothenic acid group versus placebo	2014 [27]
				Vitamins and Minerals	
150	observational study	evaluation of plasma levels of vitamin A, E and zinc in AV patients in relation to the severity of the disease *n* = 94 patients *n* = 56 control group	nd	-levels of vitamin E, vitamin A and zinc were lower among patients than in the control group -no statistically significant difference for plasma vitamin A levels between group 1 and 2 -vitamin E and zinc levels were significantly lower in group 2 than group 1	2014 [28]
				Minerals	
56	randomised prospective clinical trial	group 1 (*n* = 14): silymarin (3 × 70 mg/day) group 2 (*n* = 14): N-acetylcysteine (2 × 600 mg/day)	8 weeks	-silymarin, N-acetylcysteine and selenium with AV significantly reduced serum MDA, IL-8 levels and the number of inflammatory lesions	2012 [29]
		group 3 (*n* = 14): selenium (2 × 100 µg/day)			
				Probiotics and Prebiotics	
33	pre-experimental clinical study	probiotics	30 days	-oral probiotic trigger elevated IL-10 serum levels of AV	2019 [30]
12	proof-of-concept pilot study	FOS (fructo-oligosaccharides) and GOS (galactooligosaccharodes	3 months	-FOS/GOS supplementation in people with baseline insulin levels > 6 µUI/mL (*n* = 6) caused a significant reduction (from 7.8 to 4.3 µUI/mL) -concentration of C-peptide decreased from 2.1 to 1.6 ng/mL	2018 [31]
				Other Factors	
8197	epidemiologic investigation	association of soft drink consumption and intake of sugar from these drinks with prevalence of AV in adolescents	nd	-daily soft drink consumption increased the risk of moderate to severe AV in adolescents, mainly when sugar intake from this drink exceeded 100 g per day	2019 [32]
400	case controlled study	relationship between dietary intake of salty and spicy foods and the onset, severity and duration of AV *n* = 200 patients, *n* = 200 control group	24 h questionnaire	-patients with AV consumed higher daily amounts of sodium chloride (NaCl) than the control group -a negative correlation between the amount of NaCl in the diet of patients with AV was found -neither salty nor spicy food correlated with duration or severity of the disease	2016 [33]
80	randomised, double-blind, placebo-controlled clinical trial	-effects of a decaffeinated green tea extract (GTE) upon women with post-adolescent AV -receiving 1500 mg per day of decaffeinated GTE or cellulose for placebo group for 4 weeks *n* = 40–GTE group *n* = 40–placebo group	4 weeks	-statistically significant differences in inflammatory lesion counts distributed on the nose, periorally, and on the chin between the two groups -no significant differences between groups for total lesion counts -significant reductions in inflammatory lesions distributed on the forehead and cheek, and significant reductions in total lesion counts noticed in the GTE group -significant reductions in total cholesterol levels within the GTE group	2016 [34]

nd—no data.

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
