# Peer review of "Acne Vulgaris and Intake of Selected Dietary Nutrients—A Summary of Information"

_healthcare, 2021, doi:10.3390/healthcare9060668_

Round 1

Reviewer 1 Report

The presented manuscript is a throughout review on the topic of acne and nutrition, a very interesting subject of great interest to a broad readership.

However, there are some major revisions necessary:

-Please use the term Cutibacterium acnesinstead of the outdated terminology Propionibacterium acnesthroughout the entire manuscript.

-Clearly state inclusion/exclusion criteria for this review (in vitro/in vivo; prospective/retrospective trials; interventional/non-interventional); a PRISMA flow diagram should be included.

-Figure 2: The authors state, that there were no trials in line with the topic regarding the impact of protein. However, there are several clinical trials in the medical literature, as discussed in the “protein section” in the current review. Therefore, Figure 2 is misleading. What is the relevance of Figure 2?

-Page 1, line 36: open and closed comedones instead of blackheads

-Page 4, line 109-117; “fibre” as in prebiotics should be regarded exclusively and should be discussed in a separate section. Recently, there has been growing interest in the impact of probiotics and prebiotics on the clinical severity of acne vulgaris. It would be of great interest to also incorporate those studies in the present review, because an intake might prevent and alleviate acne lesions.

-Page 5, line 123-128 this is not the conclusion of this section. Please revise.

-Page 8, line 306 please revise “thanks to”

-The conclusion should be more precise than simply stating a” balanced diet” is recommended. According to your review, what is an optimal diet for acne patients?

Author Response

Dear Reviewer,

Thank you for your detailed study of our article and suggestions that improve the quality of our manuscript.

Below are our responses to your suggestions:

-Please use the term Cutibacterium acnesinstead of the outdated terminology Propionibacterium acnes throughout the entire manuscript.

Thank you for this suggestion. Indeed, it is now appropriate to use Cutibacterium acnes. Each phrase "Propionibacterium acnes" in the article has been replaced with "Cutibacterium acnes" as recommended.

-Clearly state inclusion/exclusion criteria for this review (in vitro/in vivo; prospective/retrospective trials; interventional/non-interventional); a PRISMA flow diagram should be included.

Thank you for this suggestion. In our publication we have included Figure 2 - a diagram like PRISMA, with custom modifications according to the literature search used.

Additionally, in the methodology, we have described in more detail what the inclusion and exclusion criteria were.

-Figure 2: The authors state, that there were no trials in line with the topic regarding the impact of protein. However, there are several clinical trials in the medical literature, as discussed in the “protein section” in the current review. Therefore, Figure 2 is misleading. What is the relevance of Figure 2?

Thank you for this suggestion. Indeed, we referenced more publications in the discussion - they were found in the acne literature search, but not in the literature review according to the search terms. In order to standardize the manuscript, in line with your suggestions and the suggestions of the second reviewer, we have compiled a table that summarizes other research results.

-Page 1, line 36: open and closed comedones instead of blackheads.

Thank you, we have changed this wording.

-Page 4, line 109-117; “fibre” as in prebiotics should be regarded exclusively and should be discussed in a separate section. Recently, there has been growing interest in the impact of probiotics and prebiotics on the clinical severity of acne vulgaris. It would be of great interest to also incorporate those studies in the present review, because an intake might prevent and alleviate acne lesions.

Thank you, as suggested, we have edited our text of the publication - we have prepared separate subsections on fiber, prebiotics and prebiotics. Additionally, we summarized these studies in Table 2.

-Page 5, line 123-128 this is not the conclusion of this section. Please revise.

Indeed, this is not a conclusion, so we have removed the confusing phrase "in conclusion".

-Page 8, line 306 please revise “thanks to”

We have reworded the "Conclusions" section - to avoid any miswording.

-The conclusion should be more precise than simply stating a” balanced diet” is recommended. According to your review, what is an optimal diet for acne patients?

Thanks for this tip. We have reworded the "Conclusion" section to be more specific.

Additionally, our manuscript has been checked for correctness in English.

Thanks again for reading and suggestions that will improve the quality of our manuscript.

Authors

Reviewer 2 Report

The authors performed a retrospective review of the clinical studies in the manuscript regarding the correlation between acne and nutrition. After an exhaustive introduction about the etiological causes of acne, the authors briefly propose the method of querying the search engines with the results obtained, differentiating in studies selected for specific dietary components and any exclusion criteria for works not included in the discussion. Subsequently, the dietary elements are treated individually by presenting the results of the available studies and the possible mechanisms proposed for the influence on acne.

The manuscript is smooth and well organized. The language is clear and comprehensive, however, some aspects regarding the purpose of the review should be clarified and the manuscript revised accordingly.

In detail, the presence of a paragraph concerning materials and methods implies that a systematic method has been applied to search for sources. If so, Table 1 should be replaced by a flow chart, following the PRISMA guidelines. Besides, the description of the search string, any filters, the types of studies included, the results obtained and the works excluded should be expanded.

If a systematic method had not been applied, the methods paragraph and table 1 should be removed and paragraph 3 should be titled as discussion and not results and discussion.

From table 1 and table 2 it emerges that only 4 studies were selected but during the discussion, other works are discussed (if I'm not mistaken, ref. 8, 12, 13, 15, 20, 21, 23, 28, 32, 34, 35, 37, 38, 39). I recommend inserting them in table 2, specifying the type of study for each item. If they have not emerged from the results obtained with the research method, they can still be references obtained from the bibliography of other articles.

I would have expected a hint on the skin-gut axis. Is it possible that no results have emerged concerning it?

There is anecdotal data on vitamin B12 and worsening of acne symptoms. In dealing with the B vitamins, I believe that at least a hint about it is necessary.

Author Response

Dear Reviewer,

Thank you for your detailed study of our article and suggestions that improve the quality of our manuscript.

Below are our responses to your suggestions:

- In detail, the presence of a paragraph concerning materials and methods implies that a systematic method has been applied to search for sources. If so, Table 1 should be replaced by a flow chart, following the PRISMA guidelines. Besides, the description of the search string, any filters, the types of studies included, the results obtained and the works excluded should be expanded.

If a systematic method had not been applied, the methods paragraph and table 1 should be removed and paragraph 3 should be titled as discussion and not results and discussion.

Thank you for this suggestion. In accordance with your comments and the comments of the first reviewer, we made the PRISMA scheme - making our own modification. In the methodology section, we have added more details about the inclusion and exclusion criteria.

- From table 1 and table 2 it emerges that only 4 studies were selected but during the discussion, other works are discussed (if I'm not mistaken, ref. 8, 12, 13, 15, 20, 21, 23, 28, 32, 34, 35, 37, 38, 39). I recommend inserting them in table 2, specifying the type of study for each item. If they have not emerged from the results obtained with the research method, they can still be references obtained from the bibliography of other articles.

Thank you for this suggestion. Indeed, the publications we discussed were not found when browsing the database and typing in the terms we mentioned, narrowed down according to the criteria described. As suggested, we have added a table summarizing these publications.

I would have expected a hint on the skin-gut axis. Is it possible that no results have emerged concerning it?

Thank you for this suggestion. To discuss this, we've added a separate section on the skin-gut axis, and discussed prebiotics and probiotics.

- There is anecdotal data on vitamin B12 and worsening of acne symptoms. In dealing with the B vitamins, I believe that at least a hint about it is necessary.

Thank you for this important suggestion. As you suggested, we discussed the effects of vitamin B12 at the end of the chapter on vitamins.

Additionally, our manuscript has been checked for correctness in English.

Thanks again for reading and suggestions that will improve the quality of our manuscript.

Authors

Round 2

Reviewer 1 Report

The manuscript has improved significantly after major revision and is an appreciated contribution to this promising field of research.

Reviewer 2 Report

I thank the authors for the revisions made on their manuscript which have certainly improved the quality of the work.
As regards the studies that emerged from the consultation of bibliographic lists, not directly emerging from the results of the query, they should, in any case, be inserted upstream among the sources in the flow chart and counted among the final works included and discussed.
Too many articles not considered due to lack of free access. Have you searched with university libraries access? Have the authors been contacted? With such a large number, I have some doubts that it can be considered a systematic review. Those 911 excluded works should first be selected based on title and abstract and the articles appropriate to the topic searched in-depth (I recommend using researchgate, kopernio, or unpaywall, allowed methods to facilitate the retrieval of articles).
The summary table is incomplete. I recommend entering the type of work for each item (RCT, cross-sectional, retrospective, prospective)
